# THE ROLE OF SHARED LABELS AND EXPERIENCES IN REPRESENTATIONAL ALIGNMENT

Kushin Mukherjee[1,2]* Siddharth Suresh[1,2,3]* Xizheng Yu[3], Gary Lupyan[1]

[1]Department of Psychology, University of Wisconsin-Madison, Madison, USA.
[2]Wisconsin Institute of Discovery, University of Wisconsin-Madison, Madison, USA.
[3]Department of Computer Science, University of Wisconsin-Madison, Madison, USA.
Corresponding email: `kmukherjee2@wisc.edu`

## ABSTRACT

Successful communication is thought to require members of a speech community to learn common mappings between words and their referents. But if one person's concept of CAR is very different from another person's, successful communication might fail despite the common mappings because different people would mean different things by the same word. Here we investigate the possibility that one source of representational alignment is language itself. We report a series of neural network simulations investigating how representational alignment changes as a function of agents having more or less similar visual experiences (overlap in "visual diet") and how it changes with exposure to category names. We find that agents with more similar visual experiences have greater representational overlap. However, the presence of category labels not only increases representational overlap, but also greatly reduces the importance of having similar visual experiences. The results suggest that ensuring representational alignment may be one of language's evolved functions

## 1 INTRODUCTION

How does one learn the meaning of new words? Some theoretical views describe the acquisition of word-meanings in terms of mapping — word-forms (e.g., the words 'car' and 'truck') are mapped onto the previously existing conceptual categories (CAR and TRUCK) (Fodor, 1975; Snedeker et al., 2004; Pinker, 1994; Bloom, 2002). An alternative account is that encountering such labels signals to the learner that there is a distinction worth learning, which causes the learner to privilege this distinction due to the benefits it confers on them within their speech community (Booth & Waxman, 2002; Waxman & Markow, 1995; Xu, 2002; Pomiechowska & Gliga, 2019; Wojcik et al., 2022; Lupyan & Lewis, 2017). Successful communication is contingent on the same words having sufficiently similar meanings for different agents. That is, the semantic representations evoked by words in natural language have to be *aligned* across different individuals. Critically, what must be aligned is not individual concepts but the structure of inter-concept relationships or the representational geometry. For example, if for one person a car is more similar to a truck than to a motorcycle, while for another it is the reverse, we might expect rather severe confusion. Thus, the structure of car-motorcycle-truck must be similar across agents to facilitate understanding.

What shapes our semantic representations such that word meanings come to be approximately shared among different individuals? One source of alignment is our shared biology. For instance, the human ear is typically sensitive to a specific range of frequencies and people have roughly similar profiles of sound discrimination Pumphrey (1950). This biological commonality ensures that when one person talks about a specific sound (e.g. a sound of a car honking) or tone within this range, another person, barring auditory impairments, will have a similar sensory experience of that sound. Common cognitive learning algorithms is another such source of alignment. Humans have common constraints on processing and inductive biases towards categorization that shape concept learning.

---

[1]*Equal contribution

For example, there is little risk of someone's meaning of a 'car' being only blue cars viewed from the side, as this would violate basic principles of human categorization Rosch & Lloyd (1978); Shepard (1994). A third source of alignment is the degree to which different individuals have life experiences that are broadly shared. While each individual's life journey is unique, there are many experiences such as growing up in similar locales, being exposed to common learning curricula, and being exposed to similar media that might help align conceptual systems. Thus, human conceptual representations are possibly aligned throughout the process of learning language and it is this alignment that makes linguistic communication possible in the first place. But another possibility is that alignment is achieved—in part—through language itself Lupyan & Bergen (2016); Casasanto (2015); Dingemanse (2017). On this view, rather than being just a device for conveying our thoughts, language provides an interface between minds Clark (1998); Gentner & Goldin-Meadow (2003); Gomila et al. (2012); Lupyan & Bergen (2016).

In a prior study, Suffill et al. (under review) tested the role of language in the conceptual alignment of novel shapes, which could be grouped into two categories based on visual features alone. To test the contributions of verbal labels distinct from perceptual learning towards conceptual alignment, they measured how similarly different participants grouped the shapes in 3 conditions — a baseline condition that relied on the similarity of participants' visual perception, a no-label condition where participants were first familiarized with the category structure of the shapes without labels, and a language condition where they were exposed to incidental nonsense labels for each category. Exposure to labels led to more categorical representations of the concepts (shapes), which in turn led to greater alignment between participants as measured by label-participants producing more similar sorts to other label-participants.

Here, we build on these findings and prior computational simulations that have hinted at the importance of language in aligning representations of visual concepts Roads & Love (2020); Steels et al. (2005). Similar to Roads & Love, we explore how learning from multiple signatures of categorical information, feedback from a labeling and a match-to-sample task, affects how the stimuli are represented and the extent to which the representations of different agents (neural networks) are aligned.

Unlike studies with human participants, simulating learning in artificial agents allows us to keep all the learning parameters constant while manipulating the prior perceptual experiences of each agent. This allows us to examine representational alignment of agents that vary in the overlap of their 'perceptual diets' and who are trained with or without category labels.

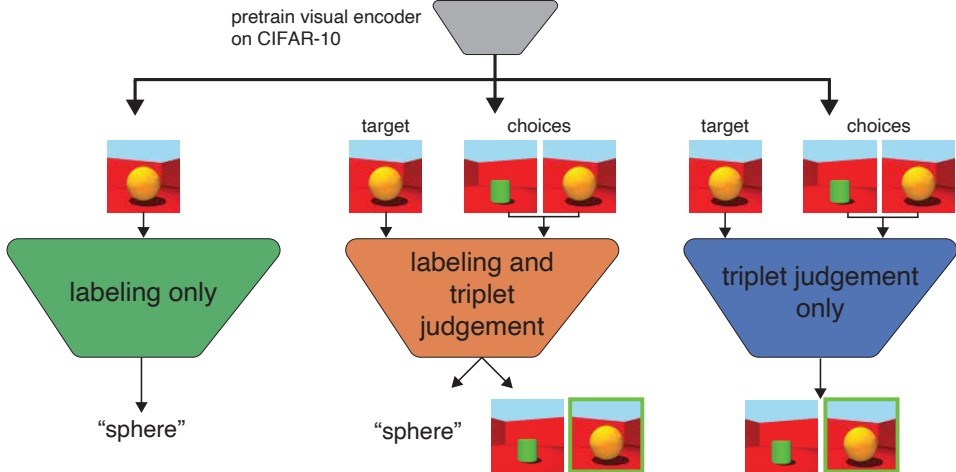

Figure 1: Overview of the 3 task conditions each pre-trained model was fine-tuned on.

## 1.1 DATASET

We leveraged two image databases for out tasks. The first was the CIFAR10 dataset consisting of 60,000 images belonging to 10 object categories. These data was used to pre-train each neural

network using the SimCLR unsupervised learning framework Chen et al. (2020) so that the networks had some prior visual knowledge (just as human participants do) before being fine-tuned on our experimental dataset and training conditions.

For our main experimental manipulations, we used the Deepmind 3D Shapes dataset Kim & Mnih (2018). This dataset consists of 480,000 rendered images of spheres, cubes, cylinders, and capsules varying in size, orientation, and color of the target image and background elements. In our experiments, we kept the color of the background elements constant to simplify learning.

We sampled from this subset of images to create 3 datasets with varying degrees of overlap with each other. Each dataset had 120 images in total with 30 images per shape category randomly sampled from the set of all possible images. We refer to the 3 datasets as **dataset A**, **dataset B**, and **dataset C**. Datasets A and B had 50% overlap in their data, datasets B and C had 33% overlap in their data, and datasets A and C had 0% overlap in their data.

## 2 METHODS

We used `PyTorch` to train neural network models to perform three tasks on the 3D shapes dataset. The three tasks were: (1) labeling the shapes of objects, (2) a match-to-sample triplet similarity judgement task analogous to that used by Suffill et al. and (3) a combination of (1) and (2). Models were first pre-trained on the CIFAR10 dataset and were then 'fine-tuned' on the 3D Shapes dataset. Models were trained on one of the three different 3D shape datasets, each of which overlapped with the remaining two to varying degrees. For example, 50% of the images in the first dataset were also present in the second dataset. This allowed us to measure alignment between two models as a function of the overlap in their training as well as whether the training included labels.

### 2.1 MODEL ARCHITECTURE AND PRE-TRAINING

Each model consisted of a simple convolutional encoder consisting of 3 convolutional layers followed by 3 linear 'dense' layers that projected to a 64-dimensional hidden layer. We pre-trained 10 variants of this encoder using the CIFAR10 dataset. Pre-training continued until the validation accuracy was greater than 85% and the mean change in accuracy across epochs was less than 2%. This ensured that all models were trained to a similar criterion before fine-tuning on the 3 task conditions.

### 2.2 TRAINING ON THE EXPERIMENTAL MATERIALS

We fine-tuned each of the 10 pre-trained models on the 3 tasks below using each of our training datasets — A, B, and C. For each pre-trained model we also fine-tuned a *second* model on dataset A so as to have 2 models that had 100% overlap in training data but different fine-tuning initializations. Thus each of the 10 pretrained models was used to further train 12 models (3 tasks × 4 datasets). 20 images were held out of each training set and used as a validation set to track network training.

**Label condition.** In this condition, a 3-layer decoder network took the latent representations from the pre-trained CIFAR10 encoder as input and was tasked with predicting the correct shape label for a given input image. This model was trained on a binary cross-entropy loss on the class logits. Each model was fine-tuned for 1000 epochs, which allowed the validation loss to stabilize.

**Triplet Judgement condition.** In this condition, the hidden layer of the pre-trained encoder projected to a single linear layer with ReLU activation. We trained a triplet loss on the output of this layer in the following way. On each iteration, 3 images would be provided to the model — a 'target' image and two 'choice' images. One of the choice images would be exactly identical to the anchor and the other option image would be a random image from one of the three other shape categories. The model's task was to guess which image matched the target image based on the cosine similarities of the latent representations. This model was trained for 1000 epochs, allowing the validation loss to stabilize.

**Label and Triplet condition.** In this condition, the pre-trained models were tasked with both providing the label for the 'anchor' image as well as performing the triplet judgement task. Both losses were equally weighted and once again the models were trained for 1000 epochs until the validation losses stabilized.

# 3 RESULTS

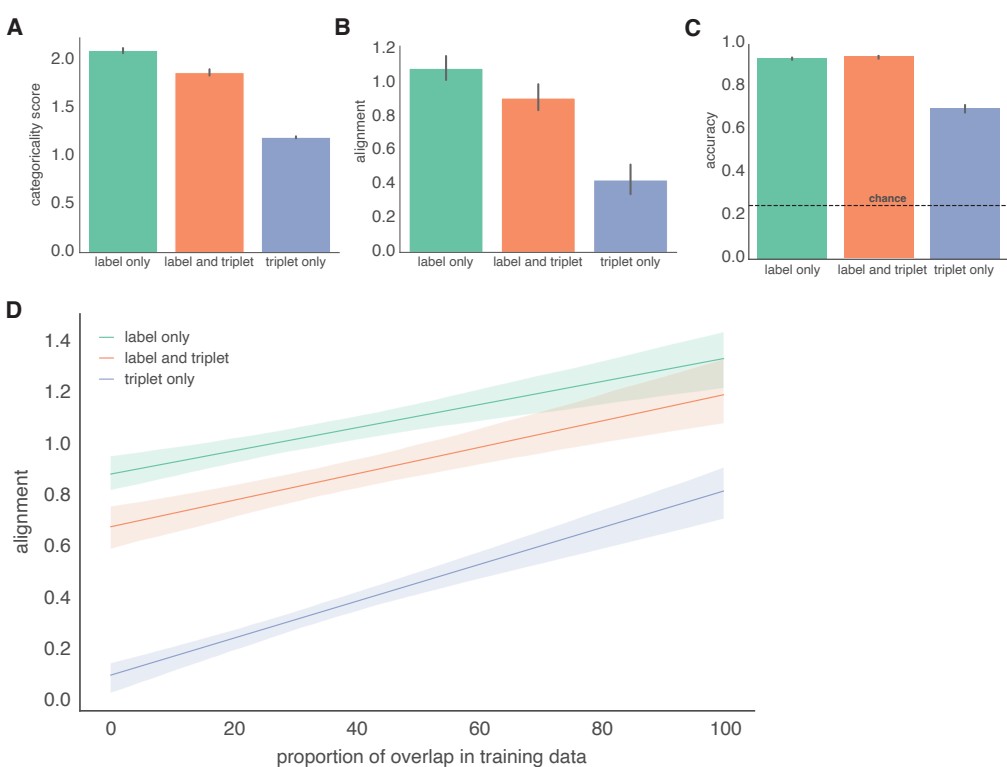

Figure 2: **(A)** Mean categoricality of learned representations, **(B)** mean conceptual alignment between pairs of models in each training condition, and **(C)** mean labeling accuracy in each training condition.**(D)** The effect of data overlap and task on representational alignment.

All results were computed with respect to a set of 480 validation images that were not shown to the networks during training. The validation set consisted of 120 images belonging to each of the 4 shape categories.

**Categoricality.** The categoricality of the learned representations is the extent to which the networks learned to represent each kind of shape as a distinct category. We quantify categoricality using the activation pattern of the encoder's final hidden layer. Categoricality is defined as follows: $Categoricality = log(\frac{distance_{between-category}}{distance_{within-category}})$, where distance refers to the cosine distance between the activation vectors, and a category consists of, e.g., all the spheres included in the validation set.

Networks trained on only the labeling task showed the greatest of categoricality ($M$=2.11, $SD$=.08) followed by networks trained on both labeling and the triplet task ($M$=1.87, $SD$=.10). The models trained on only the triplet task showed the least amount of categoricality ($M$=1.20, $SD$=.03), all $p's < .001$.

**Alignment** Alignment is a measure of how similar the representational geometry of a common set of items is across pairs of agents, i.e., neural networks. We operationalized alignment as the log-transformed multiplicative inverse of the Procrustes disparity between the activation vectors of the validation images between any given pair of networks. The more similar the representational geometry between the networks the *higher* this alignment value. Pairs of models that were trained on only labeling showed the highest alignment ($M$= 3.05, $SD$ = .73). Models trained on both tasks showed an intermediate amount of alignment ($M$= 2.58, $SD$=.79). Finally, models trained on only the triplet judgement task, i.e., with no category labels, showed the least alignment ($M$= 1.67, $SD$=.55), all $p's < .001$. As in the experiment reported by Suffill et al. (under review),

categoricality completely mediated the effect of task on alignment. When included as a predictor, the task-associated differences in alignment disappeared ($t's < 1$).

**Classification Accuracy** We also tested each model on how accurately it could classify the validation images with the correct category label. The triplet-condition models, never trained with labels, could not be expected to produce correct labels and indeed were at chance. To give these models the best possible opportunity to map their learned representations to the correct labels, we fit logistic classifiers using their activation vectors as input and the category labels as output and evaluated using 5-fold cross-validation. We took the mean accuracy on the held-out folds as the labeling accuracy. Networks trained on labels only ($M = .93$, $SD = .01$) and labeling and triplet judgements ($M = .94$, $SD = .02$) had similarly high performance. Performance of the models trained on only the triplet judgement task was much lower ($M = .70$, $SD = .05$), $p < .001$, but well above chance ($p < .001$) showing that it is possible to learn a mapping function from the network's latent states to the labels, albeit not nearly to the same level of accuracy as when the training included labels.

**Overlap in 'perceptual diets'** To test whether greater amounts of overlap in the training data led to more aligned representations and if this effect varied as function of training task, we fit a linear regression model predicting alignment from the proportion of overlap in training data, the training task, and their interaction. As clearly shown in Figure 2 D., increasing overlap led to greater alignment ($p < .001$). Even with complete overlap in perceptual experience, however, the use of labels continued to have greater alignment. Moreover, decreased overlap impacted alignment between models trained without labels significantly more than either label or label-and-triplet models ($p's < .01$).

## 4 GENERAL DISCUSSION

Despite varied life histories resulting in differential exposure to visual concepts, people generally carve the world at common joints. What supports the learning of such a shared semantics across different individuals? Recent models of the ventral visual stream suggest that computational motifs based in contrastive self-supervised learning algorithms (Zhuang et al., 2021; Konkle & Alvarez, 2022) might lead to common human-like semantic representations across agents. At the same time, computational and behavioral evidence supports the idea that language can be a vehicle of representational alignment (Roads & Love, 2020; Steels et al., 2005; Suffill et al., 2022). Here, we tested the relative contributions of both hypotheses using a simple category learning task on artificial agents. We found that training agents on labels led to representations that were more categorical relative to a condition without labels and only using a contrastive visual learning paradigm. Additionally, pairs of agents trained with labels showed more conceptual overlap relative to pairs trained without labels. The impact of task type (labeling, triplet-based contrastive, both) was mediated by this effect of categoricality, which suggests that training with labels induced more categorical representations, which in turn led to greater alignment of agents' representations.

Overall, the label-only condition led to the most categorical and aligned representations across agents without any compromises on the ability of the models to accurately label new exemplars. Our results provide computational support for the notion that language alone can greatly shape visual representations independent of visual learning processes. While our work marks an important first step, future work will seek to validate this paradigm with state-of-the-art models trained on vision-language contrastive paradigms Radford et al. (2021); Chung et al. (2022); Zhang et al. (2023) and more naturalistic learning environments. These tests will be critical for evaluating how conceptual alignment arises in the real world with naturalistic image and language input.

In summary, our results highlight the role language might play in aligning our representations of the world so as to facilitate effective communication despite sometimes vast differences in individual experiences Enfield & Kockelman (2017).

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
