# OpenReview forum: "The role of shared labels and experiences in representational alignment"
_ICLR.cc/2024/Workshop/Re-Align — ICLR 2024 Workshop Re-Align Poster_

### Official Review · Reviewer_gs5z · 2024-02-19

**Rating:** 2
**Fit:** 3
**Confidence:** 3

**Workshop Review:**

This paper explores how different visual labels (labeling only, labeling and triplet judgement, triple judgement only) impact the downstream learned representation similarity on a visual shape task.

Results illustrated that labels only elicited higher alignment and accuracy scores than the other two.

Minor: the paper is poorly written, with many typos and grammatical errors (including in the abstract).

**Reason For Not Giving Higher Score:**

The paper took a very simple idea and executed it. Nothing was technically that challenging or novel.

**Reason For Not Giving Lower Score:**

The paper still proposed an interesting experiment evaulation of different forms of labeled input.

**Reviewer Domain:**

machine learning

---

### Official Review · Reviewer_qzAn · 2024-02-23
**Strong submission that's topically aligned with the workshop**

**Rating:** 3
**Fit:** 3
**Confidence:** 2

**Workshop Review:**

Strengths:
+ very relevant to the workshop theme
+ really cool motivation: language is ambiguous and if the speaker and listener represent it differently they’ll have difficulty communication; so they look at how visual overlap, i.e. what each agent sees, influences how aligned their representation of the same language is.
+ well written and clear
+ really clearly stated controlled experiment, and well executed statistical analysis

Weaknesses:
- To be honest, I thought this was a good submission! If I had any negative comments/suggestions, I’d say that Fig. 1 would benefit from having a sphere choice that maybe isn’t identical to the target (perhaps change the view angle or the color or size or something); Fig.2 would benefit from larger text (it’s hard to see currently); lastly, you might want to cite some of these related works that study learning from triplets [1-6]
- Nit: What is CAR? If you introduce an acronym in the abstract for the first time, it’s good to spell it out.

[1] Cagatay Demiralp, Michael Bernstein, and Jeffrey Heer. 2014. Learning Perceptual
Kernels for Visualization Design. IEEE Transactions on Visualization and Computer
Graphics 20. https://doi.org/10.1109/TVCG.2014.2346978
[2] Brian McFee, Gert Lanckriet, and Tony Jebara. 2011. Learning Multi-modal
Similarity. Journal of machine learning research 12, 2 (2011).
[3] Sameer Agarwal, Josh Wills, Lawrence Cayton, Gert Lanckriet, David Kriegman,
and Serge Belongie. 2007. Generalized non-metric multidimensional scaling. In
Artificial Intelligence and Statistics. PMLR, 11–18.
[4] Ehsan Amid, Aristides Gionis, and Antti Ukkonen. 2015. A Kernel-Learning
Approach to Semi-supervised Clustering with Relative Distance Comparisons,
Vol. 9284. https://doi.org/10.1007/978-3-319-23528-8_14
[5] Andreea Bobu , Yi Liu , Rohin Shah , Daniel S. Brown , Anca D. Dragan :
SIRL: Similarity-based Implicit Representation Learning. HRI 2023: 565-574
[6] Ran Tian, Chenfeng Xu, Masayoshi Tomizuka, Jitendra Malik, Andrea Bajcsy:
What Matters to You? Towards Visual Representation Alignment for Robot Learning. CoRR abs/2310.07932 (2023)

**Reason For Not Giving Higher Score:**

N/A

**Reason For Not Giving Lower Score:**

Really well conducted, analysed, and discussed study

**Reviewer Domain:**

machine learning

---

### Official Review · Reviewer_mPHW · 2024-02-25
**relevant, clean comparisons**

**Rating:** 3
**Fit:** 3
**Confidence:** 3

**Workshop Review:**

This work is highly relevant to the workshop. It examines comparable training conditions where labeling is used or not. Labeling promotes alignment, reducing the importance of a similar visual diet. The results point to the role of language in promoting alignment.

**Reason For Not Giving Higher Score:**

NA

**Reason For Not Giving Lower Score:**

This work is highly relevant to the workshop. It examines comparable training conditions where labeling is used or not. Labeling promotes alignment, reducing the importance of a similar visual diet. The results point to the role of language in promoting alignment.

**Reviewer Domain:**

cognitive science

---

### Decision · Program_Chairs · 2024-03-02

Accept (Poster)